# Temporal Changes of Fish Diversity and Driver Factors in a National Nature Reserve, China

**DOI:** 10.3390/ani12121544

**Published:** 2022-06-14

**Authors:** Dongmei Luo, Ziwei Wan, Chenchen Jia, Yiqiu Zhu, Qin Zou, Fangfei Ye, Yan Feng, Mi Zhang, Xiaoping Wu, Xiongjun Liu, Shan Ouyang

**Affiliations:** 1School of Life Sciences, Nanchang University, Nanchang 330031, China; luodongmei07@163.com (D.L.); wzw970312@163.com (Z.W.); bb960301@163.com (C.J.); xpwu@ncu.edu.cn (X.W.); 2Aquatic Biology Protection and Rescue Center of Jiangxi Province, Nanchang 330031, China; zyq723106@163.com; 3Lushan National Nature Reserve Administration of Jiangxi Province, Nanchang 330031, China; 13970246138@139.com (Q.Z.); yefangfei0108@163.com (F.Y.); feng18970261990@sina.com (Y.F.); zhangmi_0905@163.com (M.Z.); 4Guangdong Provincial Key Laboratory of Conservation and Precision Utilization of Characteristic Agricultural Resources in Mountainous Areas, School of Life Science, Jiaying University, Meizhou 514015, China

**Keywords:** species diversity, phylogenetic diversity, homogenization, anthropogenic disturbances

## Abstract

**Simple Summary:**

Freshwater-fish diversity declined rapidly due to multiple anthropogenic disturbances. The loss of fish diversity often manifested itself in taxonomic homogenization over time. Knowledge of multi-faceted diversity (i.e., species, functional, and phylogenetic diversity) perspectives is important for biodiversity assessment and conservation planning. The results showed that the diversity of fish has declined from 2008 to 2021, with five species lost over time. We found an overall homogenization trend in the fish fauna of the study area, with a 4% increase in the taxonomic similarity among the rivers. Additionally, we found that the community structure of fish was significantly different among the rivers, and environmental filtering was the main contributor to the phylogenetic diversity of fish in 2008 and 2021. This study provides new insight into the patterns and drivers of fish-biodiversity change in the broader Yangtze River basin and informs management efforts.

**Abstract:**

Freshwater-fish diversity declined rapidly due to multiple anthropogenic disturbances. The loss of fish diversity often manifested itself in taxonomic homogenization over time. Knowledge of multi-faceted diversity (i.e., species, functional, and phylogenetic diversity) perspectives is important for biodiversity assessment and conservation planning. Here, we analyzed the change of the species diversity and phylogenetic diversity of fish in 2008 and 2021 as well as explored the driver factors of the biodiversity patterns in the Lushan National Nature Reserve. The results showed that the species diversity and phylogenetic diversity of fish have declined from 2008 to 2021, with five species lost over time. We found an overall homogenization trend in the fish fauna of the study area, with a 4% increase in taxonomic similarity among the rivers. Additionally, we found that community structure of fish was significantly different among the rivers, and environmental filtering was the main contributor to the phylogenetic diversity of fish in 2008 and 2021. This study provides new insight into the patterns and drivers of fish-biodiversity change in the broader Yangtze River basin and informs management efforts.

## 1. Introduction

Freshwater fish are one of the most diverse vertebrates and are also one of the most threatened groups globally [1,2,3]. Freshwater fish play major roles in aquatic ecosystems and are indicator organisms for detecting health of the river environment [4,5]. However, freshwater-fish biodiversity declined rapidly due to multiple anthropogenic disturbances, such as dam construction, water pollution, and overfishing [6,7,8]. More than 30% of freshwater-fish populations are declining in the current period [9]. Therefore, knowledge of fish biodiversity is crucial to inform conservation and management strategies.

Anthropogenic disturbances have put these natural environments at risk and affected biodiversity [3,8]. Assessing and quantifying the effect of anthropogenic disturbances on biodiversity is an important topic for ecologists [3,8]. Previous studies have focused on the level of species diversity (e.g., species richness, Shannon–Wiener diversity index) in assessing the effects of anthropogenic disturbances on ecological communities [10,11,12]. In contrast, study on phylogenetic diversity in assessing the effects of anthropogenic disturbances on ecological communities has not received much attention [13,14]. Species diversity reflects treating all species as equal, neglecting the differences in evolutionary relationships among species that may provide complex information of biodiversity [15,16]. Ecologists are increasingly recognizing this shortcoming, and several studies have highlighted the importance of incorporating multifaceted diversity in biodiversity management and conservation planning [17,18,19]. Recently, many studies have focused on phylogenetic diversity because of a few key reasons [19,20,21,22]. Phylogenetic diversity refers to the evolutionary relationships among species, indicating that it can reflect the evolutionary history on biological communities [18,19,20]. In addition, phylogenetic diversity emphasizes ecosystem degradation, compared with species diversity [18,19]. Therefore, knowledge of multi-faceted diversity (i.e., species, functional, and phylogenetic diversity) perspectives is important for biodiversity assessment and conservation planning [20,21,22].

Protected areas can prevent population decline of species and habitat loss, which, therefore, play an important role in protecting biodiversity and species resources [23]. The Lushan National Nature Reserve (29°30′ N–29°41′ N, 115°51′ E–116°07′ E) is located in Jiangxi Province, has abundant biodiversity, and is regarded as a key “biological refuge” in the middle and lower reaches of the Yangtze River in China [24,25]. It has also the titles of a World Heritage Site, 5A Scenic Area, and National Geopark [24,25]. The subtropical forest ecosystem, endangered wildlife species, and habitat are the primary conservation targets in this reserve [24,25]. However, fish habitats and biodiversity have been profoundly affected by human activities, such as tourism activities and water pollution (e.g., domestic sewage and tourist garbage) [25,26]. In the past, Huang et al. [27] have demonstrated the species diversity of fish in the Lushan National Nature Reserve, but few studies have explained the principal mechanisms of the interactions between the species diversity and phylogenetic diversity of fish and anthropogenic disturbances. Here, we aimed to analyze the patterns and drivers of species diversity and phylogenetic diversity of fish in 2008 and 2021 as well as to explore the driver factors of biodiversity patterns in the Lushan National Nature Reserve. This study will provide an important reference for the restoration and conservation of fish biodiversity.

## 2. Material and Methods

### 2.1. Study Area

The Lushan National Nature Reserve is an independent mountain in the middle and lower reaches of the Yangtze River, located at the confluence of the Yangtze River and Poyang Lake [26]. It covers a total area of 304.95 km^2^. Its annual average temperature is 11.4 °C, and annual precipitation is 1917 mm. The mountain stream in the reserve is radially arranged, with the streams on the east slope flowing into Poyang Lake, and the streams on the west slope flowing into Baili Lake. The substrate of the river bed is mainly sand and gravel [27].

The selection of sampling sites in this study considered the habitat variation and anthropogenic activities in the Lushan National Nature Reserve. In total, 13 streams (28 sampling sites) were chosen (Figure 1), including Taohuayuan River (THY), Guizong River (CZ), Xiufeng River (XF), Guanyinqiao River (GYQ), Haihui River (HH), Gaolong River (GL), Tongyuan River (TY), Jiandaoxia River (JDX), Lianhua River (LH), Weijia River (WJ), Shimenjian River (SMJ), Longmengou River (LMG), and Wuliqiao River (WLQ).

### 2.2. Samplings Methods

The fish-resource surveys were conducted in July and September in 2021. At each site, we first conducted interviews with local people to assess the potential presence of fish, using fish pictures from the Yangtze River. Each site was selected with similar average depths and similar capture efficiencies. Fish sampling was fully standardized using a portable electric fishing machine (CWB-2000 P, China; 12 V import, 250 V export), amounting to a total sampling area of 100 m^2^. Each site was conducted for 2 h. To enhance the species checklists at each section, we used ground cages (5 m in length, 0.5 m in height, 5 mm in mesh size), gillnets (50 m in length, 2 m in height, 1, 3, 5 cm in mesh size), and hand-held nets to collected fish samples. Fishing gear was exposed for 10 h. Fish samples were counted, measuring body length (cm) and weight (g) in the field. Live fish were released in the study area, and unidentified species were fixed in 10% formaldehyde solution and further identified in the laboratory of Nanchang University. All unidentified fish specimens were deposited at the museum in the School of Life Sciences, Nanchang University. In addition, we surveyed fish resource in the local township markets to enhance the species checklists at each sampling areas. All fish specimens were identified according to Chen (1998) [28], Chu et al. (1999) [29], and Yue (2000) [30], plus Fishbase (http://www.fishbase.org/search.php (accessed on 5 May 2022)) was used to correct the scientific names [31]. The historical fish-species presence and absence in 13 rivers of the Lushan National Nature Reserve were collected from Huang et al. (2008) [27]. Samplings methods and samplings areas of fish in 2008 were the similar to those in this study. The special fishing license required for scientific research has been obtained for this specimen collection, and the sampling has been completed with the assistance of the reserve staff and local fishers.

### 2.3. Data Analysis

The sampling completeness of fish for the study area was assessed based on abundance-based rarefaction [32]. The abundance-based rarefaction was implemented using confidence intervals (95%) in iNext online [32].

The index of relative importance (*IRI*: *IRI =* (%*N_i_* + %*W_i_*) × *f_i_*; %*N_i_* and %*W_i_* were percentage of number and percentage of weight, respectively, of species *i* in the total catches, and %*f_i_* was the occurrence frequency of species *i*) was used to measure the dominant species in the study area [33]. The relative abundance of each species at each sampling site was estimated (*P*i = *N_i_*/$\sum_{j=1}^{s}\;$*N_j_*; *S*: number of species; *N_i_* and *N_j_* were the counts of individual species in the sample). The Shannon–Weiner index (*H**:*
*H* = −∑(*P_i_*ln*P_i_*)), Margalef diversity index (*D**:*
*D* = (*S*–1)/ln*N*), Simpson dominance index (*F**: F* = 1 − ∑(*P_i_*)^2^), and Pielou evenness index (*J**:*
*J* = *H*/ln*S*) were used to analyze fish diversity and richness in each sampled section [34,35]. We analyzed the beta diversity of fish using the Sørensen dissimilarity index (*β*_sor_), spatial turnover component (*β*_sim_), and nestedness component (*β*_sne_) between the fish communities of each pair of sampling sites [36,37]. The analysis of beta diversity was performed in R [38] using the BETAPART package [39] and the VEGAN package [40]. The assemblage structure of fish was analyzed using the non-metric multidimensional scaling (NMDS) and Bray–Curtis similarity index, based on species abundance data, as performed in PRIMER 6 [41]. Before cluster analysis, the original data were converted to the fourth power to lessen the impact of extreme data on the results and bring them closer to a normal distribution. Analysis of similarity (ANOSIM) was used to determine the significance of differences in fish compositions among sampling areas, based on species abundance data, and SIMPER tests were used to determine the contributions of each fish species, based on species abundance data [42].

We downloaded the mitochondrial DNA (mtDNA) Cytb (cytochrome b) sequences of 16 fish species in this study and 17 species in 2008 from NCBI (https://www.ncbi.nlm.nih.gov/ (accessed on 18 February 2022)), the length of the sequences was a 1071 bp sequence. The phylogenetic tree was constructed using the neighbor-joining (*NJ*) method performed by Megan X. The net relatedness index (*NRI*) was used to infer the phylogenetic diversity of fish in the Lushan National Nature Reserve. *NRI* refers to the standardized effect size of the average pairwise phylogenetic distance (*MPD*_obs_) of all species actually obtained in the study area, relative to the random value of the null model (*MPD*_null_). *NRI* is the average value of the phylogenetic distance of all species in study area, and the formula was as follows [43]:$$\mathrm{NRI}=\frac{\mathrm{MPDobs}-\;\mathrm{mean}(\mathrm{MPDnull})}{\mathrm{SD}(\mathrm{MPDnull})}$$
where *MPD*_obs_ represents the observed value of the average pairwise taxonomic phylogenetic distance, based on the phylogenetic tree branch-length distance matrix, mean(*MPD*_null_) is the average of 999 random *MPD* values for each community generated by running 999 random assignments of species on the phylogenetic tree for 1000 iterations, and *SD*(*MPD*_null_) is the standard deviation of these random values. When *NRI* > 0, the species are aggregated in the phylogenetic structure, the community is composed of closely related species, and the mechanism of community construction is the environmental filtering. When *NRI* < 0, the species are divergent in the phylogenetic structure, the community is composed of distant species, and the mechanism of community construction is the competition. The *NRI* was calculated by R [38].

## 3. Results

### 3.1. Species Composition of Fish

The species accumulation curves for fish in the study area were close to asymptotic, based on relatively high sampling completeness and estimating Chao I as more than 95% of the study area (Appendix A). In 2021, a total of 1742 individuals were collected in the 13 rivers in the Lushan National Nature Reserve, including 16 species belonging to 4 orders, 8 families, and 16 genera, which was lower than those in 2008 (a total of 1320 individuals, belonging to 4 orders, 8 families, 17 genera, and 17 species; Appendix A). Cypriniformes were the most common order in 2008 and 2021 (Appendix A). Five fish species were not found in 2021 (spotted steed (*Hemibarbus maculatus*), amur catfish (*Silurus asotus*), yellow catfish (*Tachysurus fulvidraco*), small snakehead (*Channa asiatica*), and Chinese minnow (*Rhynchocypris oxycephala*)), plus four fish species were newly recorded (Bamboo fish (*Belligobio nummifer*), huang tangding (*Pseudobagrus ondon*), snakehead (*Channa argus*), and tupo fish (*Odontobutis sinensis*)). The Chinese red-listed fish species showed that 15 and 13 of the species are of Least Concern (LC) and 2 and 3 of the species are Data Deficient (DD) in 2008 and 2021, respectively (Appendix A). No Chinese red-listed fish species identified as threatened or near-threatened to extinction were recorded in 2008 and 2021 (Appendix A). All fish are species native to China in 2008 and 2021 (Appendix A).

The average body length of the 13 fish species in 2021 was lower than 10 cm, and 3 species were higher than 10 cm (*Belligobio nummifer*, *Channa argus*, Asian swamp eel (*Monopterus albus*)) (Appendix A). The average weight of the 15 fish species was lower than 50 g, and 1 species was higher than 50 g (*Channa argus*) (Appendix A). The percentage of number and weight of freshwater minnow (*Zacco platypus*), makou (*Opsariichthys bidens*), shrimp goby (*Rhinogobius giurinus*), and grouper (*Acrossocheilus parallens*) were higher than those in other species in 2021 (Appendix A). The dominant species were *Zacco platypus*, *Rhinogobius giurinus*, *Acrossocheilus parallens*, and *Opsariichthys bidens* in 2021, based on *IRI* (Appendix A).

### 3.2. Diversity of Fish

The diversity and abundance of fish in 2021 (*H* = 1.21, *D* = 2.01, *F* = 0.53, *J* = 0.44) were lower than those in 2008 (*H* = 1.72, *D* = 2.22, *F* = 0.32, *J* = 0.61; Figure 2). The diversity and abundance of fish in GYQ were higher than those in other rivers in 2021, and the diversity and abundance of fish in WLQ were higher than those in other rivers in 2008. The lowest diversity and abundance of fish were found in TY for 2021 and WJ for 2008. Beta diversity in 2021 (*β*_sor_ = 0.78) was lower than in 2008 (*β*_sor_ = 0.79). Changes in beta diversity were predominantly driven by spatial turnover (*β*_sim_ = 0.67 in 2008; *β*_sim_ = 0.62 in 2021), compared to species nestedness (*β*_sne_ = 0.12 in 2008; *β*_sne_ = 0.16 in 2021; Figure 3). Differences between 2008 and 2021 for the Sørensen dissimilarity index indicated an overall homogenization of species composition over time (Δ*β*_sor_ = 0. 02).

There were 10 rivers (THY, GZ, XF, GYQ, GL, TY, JDX, LH, SMJ, WLQ, and HH) in 2008 and 6 rivers (THY, XF, HH, GL, WJ, and WLQ) in 2021 with *NRI* > 0, indicating environmental filtering was the main contributor to the community construction of fish (Table 1). There was one river (OTM) in 2008 and six rivers (GYQ, TY, JDX, LH, SMJ, and LMG) in 2021 with *NRI* < 0, indicating competition was the main contributor to the community construction of fish (Table 1). The *NRI* of the Lushan National Nature Reserve in 2008 and 2021 was greater than zero, indicating environmental filtering was the main contributor to the community construction of fish (Table 1).

### 3.3. Community Structure of Fish

The NMDS and Bray–Curtis similarity results showed that the 13 rivers in 2008 and 2021 were both divided into three groups (Figure 4). The A group was WJ; the B group was LH, TY, GL, OTM, and GZ; and the remaining rivers were the C group in 2008. The A group was WJ, the B group was THY, GZ, GYQ, and GL, and the remaining rivers were the C group in 2021. ANOSIM results in 2008 and 2021 showed that community structure of fish among rivers were significantly different (*R*^2^ = 0.642 in 2008, *R*^2^ = 0.516 in 2021, *p* < 0.05). SIMPER tests showed that the average dissimilarity between the first group and the second group was the highest in 2021, and the average dissimilarity between the first group and the second group as well as the first group and the third group were both the highest in 2008 (Table 2). The contribution rate of *Zacco platypus* was the highest among the three groups in both 2008 and 2021 (Table 2).

## 4. Discussion

Multiple anthropogenic disturbances, such as dam construction, land use, water pollution, and overfishing, affected the diversity of fish in the freshwater ecosystems [3,8,44,45]. In this study, we found that the diversity of fish in the Lushan National Nature Reserve experienced a decline from 2008 to 2021. Five fish species were extirpated in 2021. The phylogenetic diversity of fish assemblages declined over time, which was usually recognized as a negative signal of biodiversity variation [17,18,19]. These results indicated that multiple anthropogenic disturbances have affected the composition and diversity of fish in the study area. Similar results were also reported for other large rivers in China (e.g., the Yangtze River and the Yellow River [14,46,47]) and globally [43,47]. Indeed, in the field investigation, small dams, land use, and water pollution were found in many rivers in the Lushan National Nature Reserve, resulting in the extirpation of fish in this study. Land use and water pollution was found in the Gaolong River, Guizong River, Tongyuan River, and Lianhua River, resulting in the extirpation of *Hemibarbus maculatus* and *Rhynchocypris oxycephala*. Fish assemblages are sensitive to the environmental change affected by multiple anthropogenic disturbances [7,8]. For example, the yield of Chinese paddlefish (*Psephurus gladius*) in the Yangtze River basin rapidly declined from 25 t in 1970s, with the timing of extinction likely by 2005 and no later than by 2010, due to being increasingly affected by various anthropogenic stressors [48]. Chinese sturgeon (*Acipenser sinensis*) and reeves shad (*Tenualosa reevesii*) were extirpated in the Yangtze River basin after construction of the Gezhouba Dam and the Three Gorges Dam [8]. Dam construction caused the hydrological disconnection of river networks, fragmented aquatic habitats, and changed hydrological conditions, resulting in changes of fish diversity [49,50,51]. Some consequences of damming have been highlighted to cause a decline in freshwater species richness in the river networks [10,52,53]. River ecosystems have experienced hydrological disconnection affected by dam constructions, resulting in fish fauna homogenization [8,54,55,56]. This study demonstrated an overall homogenization trend in the fish fauna of the Lushan National Nature Reserve, with a 4% increase in taxonomic similarity among rivers over time. In addition, water pollution has caused the deterioration of water quality (e.g., increased N and P concentrations) and indirectly affected fish diversity [8]. Domestic sewage and tourism garbage pollution has changed the water quality of the reserve, resulting in a decrease in the sensitive fish species in this study. Therefore, freshwater fish in this study area may be under multiple disturbances, exposing the species to greater risks, so study on the interactions between multiple stressors needs further attention [57,58,59,60].

Our results demonstrated that the community structure of fish was significantly different among rivers, and environmental filtering was more important than competition in affecting the fish diversity. Similarly, other studies also reported that environmental filtering was the main contributor to the community construction of fish [60,61]. The predominant role of environmental filtering can be attributed to the strong environmental characteristics across rivers. These rivers in the Lushan National Nature Reserve cover a wide range of habitat divergence, which further augments the importance of environmental filtering. Environmental characteristics are important for determining community structure of fish [62,63,64,65]. Several recent studies have also highlighted the deterministic role of environmental variables (e.g., nutrients) in affecting fish assemblages [66,67,68]. For example, some temperature-sensitive species showed different changes in distribution, based on bottom temperature variability [66], while some other species’ dispersal was based on changes in the ambient water temperature, and their dispersal was related to length and maturity levels [67]. The frequent discharge of warm or cold water, flow regime changes, and other environmental variables in the downstream areas affected by dams can establish a novel environmental filter, selecting species with corresponding traits for living in downstream sites [68]. Therefore, it is not surprising that environmental filtering appears to be the major determinants of fish diversity in the study area.

## 5. Conclusions

As human activities continue to transform global freshwater ecosystems, a key objective is to develop dynamic strategies for the conservation and management of fish diversity to adapt to changing environmental conditions, while maintaining a natural biogeographic pattern [62,69,70]. Multi-faceted diversity (i.e., species, functional, and phylogenetic diversity) over space and time is considered key to pinpointing the drivers of community variability, which can help inform conservation prioritization and planning [20,21,22]. Our results demonstrated that the species diversity and phylogenetic diversity of fish in the Lushan National Nature Reserve experienced a decline from 2008 to 2021, and the community structure of fish was significantly different among rivers, while environmental filtering was the main contributor to the community construction of fish. These findings provide new insight into biodiversity conservation and the restoration of river ecosystems in the reserve. Given that the fish species diversity and phylogenetic diversity indices were greatly impacted by multiple anthropogenic disturbances, we suggest related biodiversity conservation should be carried out, due to the need for environmental conservation in the reserve. We should appropriately manage small dams, land use, and water pollution, otherwise continuing environmental change will result in drastic losses of biodiversity, and local or global extinction may occur [3,8,44,45].

## Figures and Tables

**Figure 1 animals-12-01544-f001:**
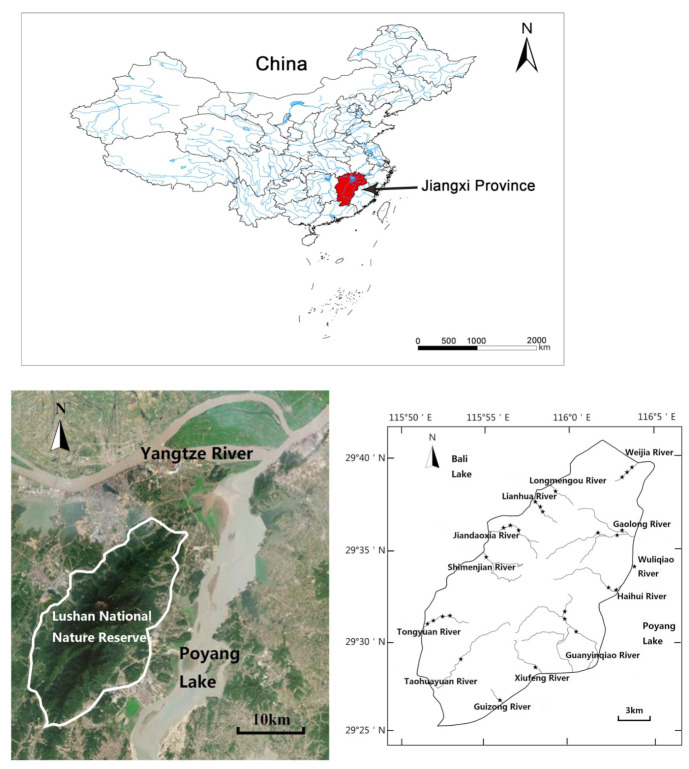
Map showing the study area of the Lushan National Nature Reserve. Black stars represent the sampling sites.

**Figure 2 animals-12-01544-f002:**
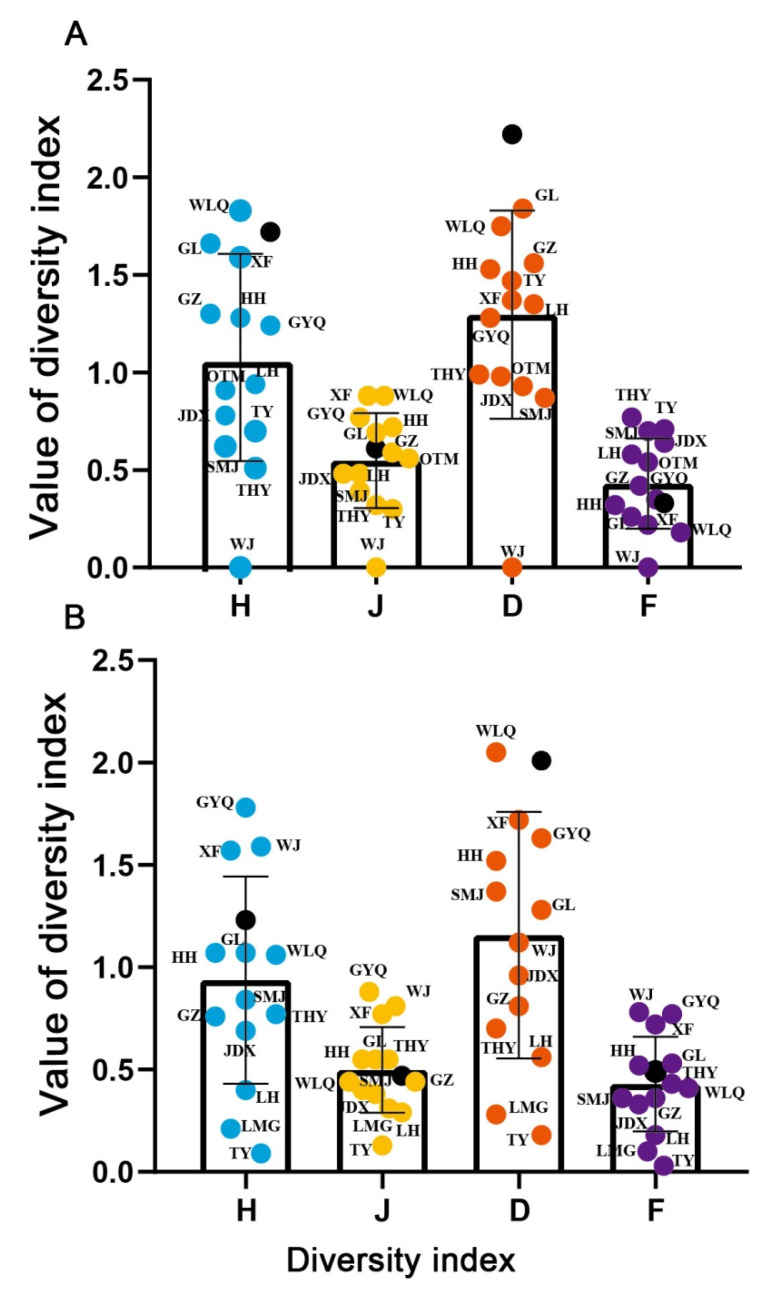
Temporal and spatial change in the diversity of fish in the Lushan National Nature Reserve in 2008 (**A**) and 2021 (**B**). River codes are the same as in Table 1. Blue, orange, red, and purple circles represent the *H*, *J*, *D*, and *F*, respectively. *H*: Shannon–Weiner index; *D*: Margalef diversity index; *F*: Simpson dominance index; *J*: Pielou evenness index.

**Figure 3 animals-12-01544-f003:**
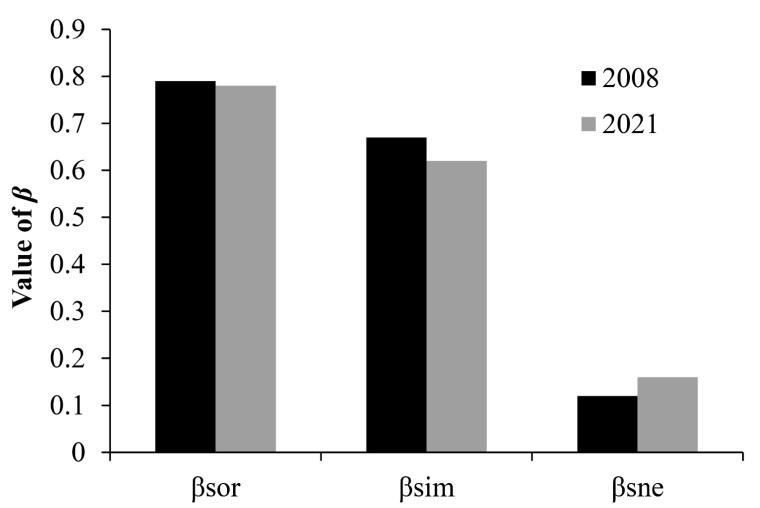
Fish species compositional dissimilarity for 2008 and 2021, quantified by the Sørensen dissimilarity index (*β*_sor_), its spatial turnover component (*β*_sim_), and its nestedness component (*β*_sne_), in the Lushan National Nature Reserve.

**Figure 4 animals-12-01544-f004:**
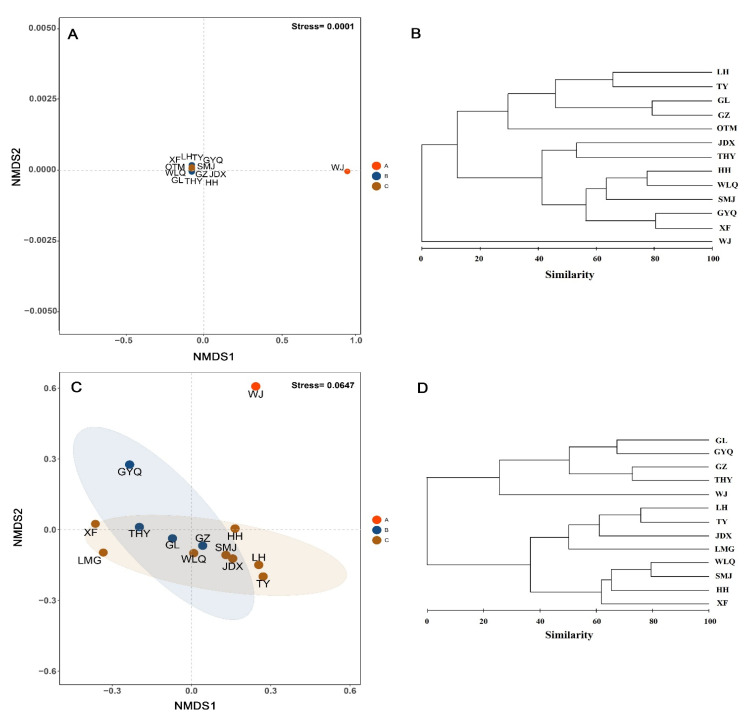
The Bray–Curtis resemblance matrix and the non-metric multidimensional scaling (NMDS) ordination for the community structure of fish in 2008 (**A**,**B**) and 2021 (**C**,**D**) in Lushan National Nature Reserve. River codes are the same as in Table 1.

**Table 1 animals-12-01544-t001:** The phylogenetic distance (*MPD*) randomization and observed, standard deviation of random values (*SD*) and net relatedness index (*NRI*) of fish in the Lushan National Nature Reserve in 2008 and 2021.

River	Codes	2008				2021			
		*MPD* Randomization	*MPD* Observed	*SD*	*NRI*	*MPD* Randomization	*MPD* Observed	*SD*	*NRI*
Taohuayuan	THY	0.061	0.060	0.008	0.173	0.122	0.109	0.019	0.658
Guizong	GZ	0.159	0.149	0.017	0.587	0.102	0.101	0.014	0.076
Xiufeng	XF	0.210	0.189	0.021	0.980	0.200	0.196	0.020	0.230
Guanyinqiao	GYQ	0.172	0.155	0.020	0.819	0.220	0.223	0.021	−0.145
Gaolong	GL	0.201	0.176	0.018	1.334	0.137	0.123	0.017	0.812
Tongyuan	TY	0.080	0.063	0.010	1.694	0.010	0.011	0.002	−0.554
Jiandaoxia	JDX	0.098	0.096	0.012	0.180	0.091	0.098	0.013	−0.536
Lianhua	LH	0.115	0.114	0.014	0.115	0.051	0.056	0.008	−0.676
Weijia	WJ	NA	NA	NA	NA	0.215	0.210	0.022	0.211
Shimenjian	SMJ	0.083	0.076	0.011	0.615	0.101	0.106	0.013	−0.426
Wuliqiao	WLQ	0.222	0.207	0.019	0.837	0.116	0.112	0.013	0.285
Haihui	HH	0.178	0.164	0.022	0.624	0.146	0.119	0.019	1.372
Shanshang	OTM	0.125	0.138	0.017	−0.823	—	—	—	—
Longmengou	LMG	—	—	—	—	0.029	0.031	0.006	−0.337
	Total	0.188	0.171	0.017	1.002	0.131	0.124	0.014	0.510

**Table 2 animals-12-01544-t002:** Dissimilarity analysis (SIMPER) between fish assemblages in the Lushan National Nature Reserve in 2008 and 2021.

2008						
Species	A group–B group (average dissimilarity = 100)		A group–C group (average dissimilarity = 100)		B group–C group (average dissimilarity = 78.12)	
	Average dissimilarity	Contribution rate (%)	Average dissimilarity	Contribution rate (%)	Average dissimilarity	Contribution rate (%)
*Zacco platypus*	46.51	46.51	45.16	45.16	33.45	42.82
*Rhynchocypris oxycephala*	11.41	11.41	—	—	9.08	11.63
*Acrossocheilus parallens*	4.74	4.74	14.16	14.16	5.11	6.54
*Rhinogobius giurinus*	15.56	15.56	—	—	8.50	10.89
*Rhodeus ocellatus*	—	—	14.51	14.51	5.57	7.13
*Misgurnus anguillicaudatus*	8.31	8.31	3.77	3.77	4.68	5.99
*Hemibarbus maculatus*	3.40	3.40	7.14	7.14	3.59	4.59
*Opsariichthys bidens*	3.26	3.26	—	—	2.95	3.78
*Liobagrus anguillicauda*	—	—	5.46	5.46	—	—
**2021**						
Species	A group–B group (average dissimilarity = 87.2)		A group–C group (average dissimilarity = 86.74)		B group–C group (average dissimilarity = 49.94)	
	Average dissimilarity	Contribution rate (%)	Average dissimilarity	Contribution rate (%)	Average dissimilarity	Contribution rate (%)
*Zacco platypus*	34.57	39.64	47.43	54.68	26.73	53.53
*Opsariichthys bidens*	14.63	16.77	11.70	13.49	2.35	4.71
*Acrossocheilus parallens*	10.45	11.99	—	—	7.99	16.01
*Rhodeus ocellatus*	7.94	9.10	8.41	9.70	—	—
*Rhinogobius giurinus*	7.20	8.26	4.76	5.49	2.78	5.57
*Misgurnus anguillicaudatus*	5.33	6.11	6.79	7.83	1.84	3.69
*Liobagrus anguillicauda*	—	—	—	—	1.48	2.96
*Monopterus albus*	—	—	—	—	1.23	2.47
*Odontobutis sinensis*	—	—	—	—	1.19	2.39

## Data Availability

All data included in this study are available upon request by contact with the corresponding author.

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
