# Peer review of "Temporal Changes of Fish Diversity and Driver Factors in a National Nature Reserve, China"

_animals, 2022, doi:10.3390/ani12121544_

Round 1
Reviewer 1 Report
Most of the corrections are given in the pdf of the manuscript. Besides that, two remarks:
- give the professional/native speaker to revise the text as there are some writing errors, and
- leave out the titles of subchapters in the discussion, especially as they do not follow those in the Results.

Author Response
Manuscript ID animals-1755639
Temporal changes of fish diversity and driver factors in a national nature reserve, China
Dear Reviewer 1,
Thank you for your thoughtful and constructive of our manuscript. We have made revisions to the manuscript as suggested by you. Below are the responses to the comments. The responses are in red text.
Most of the corrections are given in the pdf of the manuscript. Besides that, two remarks: give the professional/native speaker to revise the text as there are some writing errors, and leave out the titles of subchapters in the discussion, especially as they do not follow those in the Results.
Reply: We have revised the view according the pdf of the manuscript. In addition, we give the professional speaker to revise the manuscript as there are some writing errors, and leave out the titles of subchapters in the discussion.

Reviewer 2 Report
This paper is useful, but needs some additions to be valuable.
1) Please include common fish names in addition to scientific names.
2) English editing needed throughout.
3) Discussion is too short to explain the results. With 5 fish species being extirpated, I would like to see details of where pollution and dams occurred vs locations of these fish losses. More comparisons to extirpations in other locations globally are needed.
4) In the methods, a number of methods for collecting fish species was included. In the results, all fish species are lumped regardless of how data on them was collected (surveys vs nets vs fish markets). I think you need to compare the completeness of these sampling methods in addition to simply lumping them. How much species overlap was there?
5) Figure 2: Include y axis legend.
6) Figure 4: Somewhere you need to explain why site WJ is so different from the others.
Author Response
Manuscript ID animals-1755639
Temporal changes of fish diversity and driver factors in a national nature reserve, China
Dear Reviewer 2,
Thank you for your thoughtful and constructive of our manuscript. We have made revisions to the manuscript as suggested by you. Below are the responses to the comments. The responses are in red text.
This paper is useful, but needs some additions to be valuable.
1) Please include common fish names in addition to scientific names.
Reply: This was revised.
2) English editing needed throughout.
Reply: We give the professional speaker to revise the manuscript English.
3) Discussion is too short to explain the results. With 5 fish species being extirpated, I would like to see details of where pollution and dams occurred vs locations of these fish losses. More comparisons to extirpations in other locations globally are needed.
Reply: This was revised in the Discussion.
4) In the methods, a number of methods for collecting fish species was included. In the results, all fish species are lumped regardless of how data on them was collected (surveys vs nets vs fish markets). I think you need to compare the completeness of these sampling methods in addition to simply lumping them. How much species overlap was there?
Reply: To enhance the species checklists at each section, we used alternative fishing techniques to collected fish samples. In addition, we surveyed fish resource in the local township markets to enhance the species checklists at each sampling areas.
5) Figure 2: Include y axis legend.
Reply: This was revised.
6) Figure 4: Somewhere you need to explain why site WJ is so different from the others.
Reply: Our results demonstrated that community structure of fish was significantly different among rivers, and environmental filtering was more important than competition in affecting the fish diversity. Similarly, results were other studies also reported that environmental filtering was the main contributor to the community construction. The predominant role of environmental filtering can be attributed to the strong environmental characteristics across rivers. These rivers in the Lushan National Nature Reserve cover a wide range of habitat divergence, which further augments the importance of environmental filtering. Environmental characteristics are important for determining community structure of fish. This explains why WJ is so different from the others.

Reviewer 3 Report
This paper is a study of the decreasing biodiversity of freshwater fish in a Chinese inland nature reserve adjacent to the Yangtze River during 2008-2021. One innovative feature of this study is that the authors examine not just the number of diverse species (species diversity) but also their ecological significance or place in the evolution and sustainability of the ecosystem (phylogenetic diversity). The authors found that five species had disappeared and many other species had reduced in abundance between 2008 and 2021. They suggested these declines are linked to environmental filtering exacerbated by dam construction, land use changes and water pollution from domestic sewage and tourism garbage.
In my view, this paper is a creditable study of an important issue, and deserves to be published. However, I am a social scientist not a natural scientist, so I do not possess the expertise to evaluate the scientific credentials of the methodology used by the authors. There is, nevertheless, one methodological issue which I would like the authors to consider in relation to the cause(s) of species decline. They say that environmental filtering was the main contributor to species decline and they also allude to several anthropocentric impacts. Are these anthropocentric impacts part of environmental filtering or are they independent causes of species decline? Moreover, what evidence have the authors obtained to enable them to make these assertions about the causes of species decline? Finally, could the authors explain more explicitly what their results tell them not only of the decline of species numbers (species diversity) which they describe very clearly, but of the ecological significance of this decline (phylogenetic diversity) which they do not describe so clearly?.
Finally, could the authors explain more explicitly what their results tell them not only of the decline of species numbers (species diversity) which they describe very clearly, but of the ecological significance of this decline (phylogenetic diversity) which they do not describe so clearly.
Author Response
Manuscript ID animals-1755639
Temporal changes of fish diversity and driver factors in a national nature reserve, China
Dear Reviewer 3,
Thank you for your thoughtful and constructive of our manuscript. We have made revisions to the manuscript as suggested by you. Below are the responses to the comments. The responses are in red text.
This paper is a study of the decreasing biodiversity of freshwater fish in a Chinese inland nature reserve adjacent to the Yangtze River during 2008-2021. One innovative feature of this study is that the authors examine not just the number of diverse species (species diversity) but also their ecological significance or place in the evolution and sustainability of the ecosystem (phylogenetic diversity). The authors found that five species had disappeared and many other species had reduced in abundance between 2008 and 2021. They suggested these declines are linked to environmental filtering exacerbated by dam construction, land use changes and water pollution from domestic sewage and tourism garbage.
In my view, this paper is a creditable study of an important issue, and deserves to be published. However, I am a social scientist not a natural scientist, so I do not possess the expertise to evaluate the scientific credentials of the methodology used by the authors. There is, nevertheless, one methodological issue which I would like the authors to consider in relation to the cause(s) of species decline. They say that environmental filtering was the main contributor to species decline and they also allude to several anthropocentric impacts. Are these anthropocentric impacts part of environmental filtering or are they independent causes of species decline? Moreover, what evidence have the authors obtained to enable them to make these assertions about the causes of species decline? Finally, could the authors explain more explicitly what their results tell them not only of the decline of species numbers (species diversity) which they describe very clearly, but of the ecological significance of this decline (phylogenetic diversity) which they do not describe so clearly?.
Reply: The impact of human activities is both the cause of species decline and part of environmental filtering. Species reduction was used to determine the species composition of the reserve from the perspective of species diversity, while environmental filtration was used to analyze the driving factors of fish community construction from the perspective of phylogenetic diversity. When species diversity decreased, we speculated that these human activities in the reserve affected the species composition according to the field survey results and related literature research. Then the phylogenetic diversity method was used to study and verify. The principle of phylogenetic community structure holds that when the living environment conditions of organisms are relatively special, the community composition is close relatives that adapt to the environment, and environmental filtering dominates the community construction. When the environmental pressure of organisms is small, the overlap of food causes the dispersion of related species, and more distant species are accommodated in the community, competition dominates the community construction.

Reviewer 4 Report
Manuscript ID animals-1755639
Temporal changes of fish diversity and driver factors in a national nature reserve, China
General comments:
The manuscript analyzed the change of species diversity and phylogenetic diversity of fish in 2008 and 2021, and to explore driver factors of biodiversity patterns in the Lushan National Nature Reserve, China. The results showed that diversity of fish have declined, where five species were lost over time. The theme is quite important because it deals with fish species diversity in a Nature Reserve. The manuscript meets the scope of the journal, however it requires deep adjustments and complementation so that it can be accepted for publication. My considerations are commented out in text boxes, throughout the pdf file, which was attached in the submission of this review. Many important considerations are, in the Introduction, which needs to be expanded and deepened, especially the justifications and in the approach of legislation dealing with Natural Reserves, both local and global. The major problem in the manuscript is in Material and Methods: this topic that most needs to be complemented, because there is a lack of information, necessary to give credibility to the results obtained. The Discussion is poor and needs to be better explored, in order to complement the important approach that the objectives propose. The authors should explore the legislation and what suggestions they would make, in order to reduce anthropic impacts, seeking to improve the conservation of ichthyofauna.

Author Response
Manuscript ID animals-1755639
Temporal changes of fish diversity and driver factors in a national nature reserve, China
Dear Reviewer 4,
Thank you for your thoughtful and constructive of our manuscript. We have made revisions to the manuscript as suggested by you. Below are the responses to the comments. The responses are in red text.
The manuscript analyzed the change of species diversity and phylogenetic diversity of fish in 2008 and 2021, and to explore driver factors of biodiversity patterns in the Lushan National Nature Reserve, China. The results showed that diversity of fish have declined, where five species were lost over time. The theme is quite important because it deals with fish species diversity in a Nature Reserve. The manuscript meets the scope of the journal, however it requires deep adjustments and complementation so that it can be accepted for publication. My considerations are commented out in text boxes, throughout the pdf file, which was attached in the submission of this review. Many important considerations are, in the Introduction, which needs to be expanded and deepened, especially the justifications and in the approach of legislation dealing with Natural Reserves, both local and global. The major problem in the manuscript is in Material and Methods: this topic that most needs to be complemented, because there is a lack of information, necessary to give credibility to the results obtained. The Discussion is poor and needs to be better explored, in order to complement the important approach that the objectives propose. The authors should explore the legislation and what suggestions they would make, in order to reduce anthropic impacts, seeking to improve the conservation of ichthyofauna.
Reply: We have revised the view according the pdf of the manuscript. The special fishing license required for scientific research has been obtained for this specimen collection, and the sampling has been completed with the assistance of the reserve staff and local fishermen. We revised the view in Material and Methods according to pdf file. We also revised and added the view in the Discussion according to pdf file.

Round 2
Reviewer 4 Report
Dears,
The authors partially carried out the required revisions. Few additions were made to the Introduction and Discussion, which should have been enriched. The critical point for the manuscript to become robust and replicable was the lack of standardization of the capture effort for the calculation of abundance. Considering that different fishing gears were used, the authors must calculate the catch per standardized effort unit (Gibson-Reinemer et al., 2016, https://doi.org/10.1139/cjfas-2016-0003), to obtain the abundance data (not use fish numbers) by species/location.
Data analysis must be done again from these abundance calculations. Without accomplishing this, the manuscript lacks data credibility.
Author Response
Manuscript ID animals-1755639
Temporal changes of fish diversity and driver factors in a national nature reserve, China
Dear Reviewer 4,
Thank you for your thoughtful and constructive of our manuscript. We have made revisions to the manuscript as suggested by you. Below are the responses to the comments. The responses are in red text.
The authors partially carried out the required revisions. Few additions were made to the Introduction and Discussion, which should have been enriched. The critical point for the manuscript to become robust and replicable was the lack of standardization of the capture effort for the calculation of abundance. Considering that different fishing gears were used, the authors must calculate the catch per standardized effort unit (Gibson-Reinemer et al., 2016, https://doi.org/10.1139/cjfas-2016-0003), to obtain the abundance data (not use fish numbers) by species/location.
Data analysis must be done again from these abundance calculations. Without accomplishing this, the manuscript lacks data credibility.
Reply:
- Thank you for your thoughtful and constructive of our manuscript. We revised again the view in the Introduction and Discussion according to your first comments.
- To enhance the species checklists at each section, we used alternative fishing techniques to collected fish samples. In addition, we surveyed fish resource in the local township markets to enhance the species checklists at each sampling areas. These, these different fishing gears added species checklists, and no abundance data.
- Some data analysis done from abundance data calculations (e.g. relative abundance, Shannon-Weiner index, Simpson dominance index, Pielou evenness index). The index of relative importance (IRI) based on number percentage, weight percentage, and frequency of occurrence was used to measure fish dominance in catches and calculated as follows: IRIi = (%Ni+%Wi) × fi, where %Ni and %Wi were percentage number and percentage weight, respectively, of species i in the total catches, and %fi was the occurrence frequency of species i.
- The relative abundance of each species at each sampling site was estimated by: Pi = Ni /Nj, where S= number of species and Ni and Nj were the counts of individual species in the sample. The Shannon-Weiner index (H: H =–∑(PilnPi)), Margalef diversity index (D: D =(S–1)/lnN), Simpson dominance index (F: F =1–∑(Pi)2 ) and Pielou evenness index (J: J =H/lnS) were used to analyze fish diversity and richness in each sampled section, where S = the total number of species in each sample collected in the river.
- Baselga (2010) systematically proposes the beta diversity decomposition method based on the Sørensen index (βsor), which is decomposed into spatial turnover component (βsim) and nestedness component (βsne). Sørensen index ranges from 0 to 1, representing that no species and all species are common among the two areas, respectively.
- We performed a Bray-Curtis resemblance matrix based on species abundance data per section. Before cluster analysis, the original data were converted to the fourth power to lessen the impact of extreme data on the results and bring them closer to a normal distribution. The resemblance matrix was used to create a two-dimensional, non-metric multidimensional scaling (NMDS) plot. Additionally, similarities were analyzed by a hierarchical cluster analysis using group means. We used ANOSIM tests to assess assemblage similarity between sampling sites based on species abundance data. SIMPER tests were used to determine the contributions of each species to any differences based on species abundance data.
- We downloaded the mitochondrial DNA (mtDNA) Cytb (cytochrome b) sequences of 16 fish species in this study and 17 species in 2008 from NCBI (https://www.ncbi.nlm.nih.gov/), the length of sequences were 1071bp sequence. The phylogenetic tree was constructed using the neighbor-joining (NJ) method performed by Megan X. The net relatedness index (NRI) was used to infer the phylogenetic diversity of fish in the Lushan National Nature Reserve. NRI refers to the standardized effect size of the average pairwise phylogenetic distance (MPDobs) of all species actually obtained in study area relative to the random value of the null model (MPDnull). NRI is the average value of the phylogenetic distance of all species in study area.
Therefore, we revised these view in data analysis according to your comments.
